# Physical-Chemical and Nutritional Characterization of Somali *Laxoox* Flatbread and Comparison with Yemeni *Lahoh* Flatbread

**DOI:** 10.3390/foods12163050

**Published:** 2023-08-14

**Authors:** Antonella Pasqualone, Francesca Vurro, Erin Wolgamuth, Salwa Yusuf, Giacomo Squeo, Davide De Angelis, Carmine Summo

**Affiliations:** 1Food Science and Technology Unit, Department of Soil, Plant and Food Science (DISSPA), University of Bari ‘Aldo Moro’, Via Amendola 165/a, 70126 Bari, Italy; francesca.vurro@uniba.it (F.V.); giacomo.squeo@uniba.it (G.S.); davide.deangelis@uniba.it (D.D.A.); carmine.summo@uniba.it (C.S.); 2Brussels Institute of Advanced Studies (BrIAS) Fellow 2022/23, Elsene, 1050 Brussels, Belgium; 3Independent Researcher, MA, Dubai Investment Park 1, Dubai, United Arab Emirates; 4Independent Researcher, BA, Hargeisa, Somalia; salwacali08@gmail.com

**Keywords:** sorghum, bioactive compounds, artisan food, ethnic food, food diversity

## Abstract

The physical–chemical and nutritional characteristics of Somali *laxoox* and Yemeni *lahoh* flatbreads have not been studied to date, nor have their possible similarities been investigated. Fieldwork was carried out in Somaliland (northwest Somalia), at nine different households, to collect Somali and Yemeni flatbreads. The nutritional characteristics (12.47–15.94 g/100 g proteins, 2.47–4.11 g/100 g lipids) and the total phenolic compounds (5.02–7.11 mg gallic acid equivalents/g on dry matter) were influenced by the natural variability of manual food preparation, as well as variability in the recipes used. All the breads had a porous structure. Cell density varied from 22.4 to 57.4 cells/cm^2^ in the Somali flatbreads, reaching 145 cells/cm^2^ in the Yemeni flatbreads. Higher amounts of refined flour increased the pale color of the breads. The principal component analysis highlighted differences between Yemeni and Somali flatbreads, pointing out a certain variability within the latter, with two samples forming a separate subgroup.

## 1. Introduction

Flatbreads have ancient origins and play a fundamental role in rural food culture in many parts of the world [1]. Somali traditional fermented flatbread, with a slightly sour flavor and a typically spongy consistency, is known as “*laxoox*” in northern and western regions (Somaliland) and “*canjeero*” in southern and eastern regions of Somalia. This flatbread is strongly related to Somali culture; the familiar sound of vigorous manual batter mixing every evening has been likened to a lullaby in popular culture, linked to a sense of security that bread will be available the following morning [2]. Despite its local ubiquity, this bread has been little investigated, with the exception of an in-field study which documented its formulation, production methods, and consumption patterns in various cities in Somalia, Somaliland, and Ethiopia’s Somali State [3]. The production of *laxoox*, in detail, follows four “styles” which have been denominated as “heritage”, “new heritage”, “innovative”, and “global” [3]. Heritage production (featuring hand-ground whole grains with little or no wheat; handmade pregelatinized dough; and spontaneous fermentation or the addition of a starter from the previous baking batch) is linked to an historic era of pastoralism, while the other three styles (featuring combinations of commercial wheat flour and/or commercial pregelatinized dough and/or commercial baker’s yeast) originated in the Somali civil conflict and continue today [3].

As their similar names suggest, Somali *laxoox* may be related to Yemeni *lahoh*. The proximity of Somalia and Yemen across the Gulf of Aden has enabled trade and migration in both directions for centuries. The earliest documented evidence of Somali diaspora in Yemen dates to a neighborhood plan in 18th century Mokha [4]. According to the 1870 logs of F.M. Hunter, who supervised the 1987 Aden census, Somali men were often merchants who traded between Yemen and Somaliland, while many women prepared and sold food and drink, including “cakes of fermented and unfermented bread”, such as *laxoox* [4]. Later, a vast Somali diaspora emigrated to several countries, including Yemen, after the collapse of Siad Barre’s regime in January 1991 and during the subsequent civil war. However, as a result of the successive fighting in Yemen, many of the hundreds of thousands of Somali people to whom Yemen offered refugee status over the decades were forced to return. More recently, Yemeni conflict refugees have increasingly crossed the Gulf of Aden to Somaliland and other states in the Horn of Africa [5].

All foods have a role beyond their practical and nutritional functions, but bread carries the greatest symbolic significance [6]. Bread has an important cultural value such that scarcity or price increases tend to have significant social repercussions. Preferences for food, including bread, are influenced by family habits, traditions, religion, and income. In the context of repeated displacement, food in general, and bread in particular, play essential roles in nourishing both the body and soul, physically sustaining people while representing, in a word, “home”.

Despite the social, cultural, and nutritional importance of these flatbreads, their physical–chemical and nutritional characteristics have not been studied thus far, nor have the possible similarities among them been investigated. Therefore, the aims of this work were to carry out nutritional and chemical characterization of Somali *laxoox* and define the physical features related to its appearance, in comparison with samples of Yemeni *lahoh* flatbread, to ascertain the differences and similarities between them.

## 2. Materials and Methods

### 2.1. Collection of Bread and Flour Samples

Bread sample collection was carried out in Somaliland (northwest Somalia) in September 2022, at nine different households. Each household provided a baking batch from which three breads were collected to account for the variability of manual preparation, for a total of 27 samples. Seven baking batches (coded A and D−I) were Somali *laxoox* breads and two (coded B and C) were Yemeni *lahoh* breads. The Yemeni samples were collected at two Yemeni refugee households living in Somaliland. One household provided the *lahoh Ariiqy* bread (meaning “*lahoh* bread from Aruuq Village”), hand-delivered by relatives still living in Aruuq Village, near Tais (Yemen). The other household, whose members originated in Sana’a (Yemen), prepared the ordinary Yemeni *lahoh* sample, *Sana’ani* style.

Flour mixtures corresponding to four baking batches (E, F, G, I) were also collected. These mixtures were prepared by the households by mixing refined wheat flour with other flours produced from various unpackaged (bulk) grains purchased and milled at local open-air markets. These whole grain blends, known as *budo*, are either specific to the dietary and taste preferences of the buyer and her household or are customized by retail grain saleswomen in the markets. The *budo* flour blend corresponding to baking batch A was a pre-packaged commercial sample solely composed of red sorghum.

The composition of each *budo* flour blend (including the non-sampled ones) and the ratio of supplementation with refined wheat flour are reported in Table 1, along with the other ingredients used.

The production method of Somali breads, described in detail in a previous study [3], involves the preparation of a batter composed of flour ingredients, water, and a portion of the batter from the precedent breadmaking, acting as microbial starter; followed by leavening overnight; pouring a thin layer of the fermented batter with a spiral motion onto a greased pan placed either on a firebox, a gas stove, or a wood fire on the ground; and finally baking for 2–5 min under a domed lid, without flipping.

### 2.2. Determination of the Nutritional Composition

The protein (N × 5.7) content of the flour and breads was determined as described by the AACC method 46–11.02 [7]. The lipid fraction of flours was determined by Soxhlet extraction, using diethyl ether as the solvent, with a semi-automatic extraction system (Velp Scientifica srl, Usmate, Italy) according to the AOAC method 945.38 F [8]. The lipid fraction of flatbreads was extracted and determined as described by the AOAC method 922.06 [8]. The total dietary fibers of flour and breads were determined by the enzymatic–gravimetric procedure, according to the AOAC method 991.43 [8]. The available carbohydrates were calculated by difference. The results were expressed as g/100 g of dry matter (d.m.). Three replicates were carried out for each determination.

### 2.3. Determination of the Content of Total Bioactive Compounds

The extraction of the total phenolic compounds was conducted with methanol:water 80:20 *v*/*v*. Samples (1 g) were added to 8 mL of extracting solvent and kept in an ultrasound bath (CEIA international S.A., Viciomaggio, Italy) for 15 min at room temperature, then shaken for 30 min and centrifuged (Thermo Fisher Scientific, Osterode am Harz, Germany) for 10 min at 8000× *g* at 4 °C. The extracts were subjected to the Folin–Ciocalteu reaction as described in Pasqualone et al. (2014) [9], then the phenolic compounds were spectrophotometrically quantified at 765 nm by a Cary 60 UV–Vis spectrophotometer (Agilent Technologies, Santa Clara, CA, USA) and expressed as gallic acid equivalents (Sigma-Aldrich, Saint Louis, MO, USA). The analysis was carried out in triplicate.

The total carotenoid pigments were extracted with water-saturated *n*-butyl alcohol as described in Pasqualone et al. (2013) [10], measuring the absorbance of the extracts at 435.8 nm by a Cary 60 UV–Vis spectrophotometer (Agilent Technologies, Santa Clara, CA, USA). The total carotenoid content was expressed as β-carotene, considering the extinction coefficient of 1.6632 for a solution of 1 mg of β-carotene in 100 mL of water-saturated *n*-butyl alcohol, according to the AACC method 14–50.01. The analysis was carried out in triplicate.

The extraction of the total anthocyanin compounds was carried out using methanol:water:formic acid 80:18:2 *v*/*v*/*v*, according to Troilo et al. (2022) [11], then the absorbance of the solution was determined at 535 nm by a Cary 60 UV–Vis spectrophotometer (Agilent Technologies, Santa Clara, CA, USA). The total anthocyanin compounds were expressed as cyanidin 3-*O*-glucoside (Phytoplan, Heidelberg, Germany). The analysis was carried out in triplicate.

### 2.4. Determination of the Antioxidant Activity

The antioxidant activity was determined by assessing the ability of the samples to reduce the 2,2-diphenyl-1-picrylhydrazyl radical (DPPH•, CAS 1898-66-4, Sigma-Aldrich, Saint Louis, MO, USA), absorbing at 515 nm, to diphenyl picryl hydrazine (not absorbing at that wavelength) due to the presence of hydrogen-donating antioxidants. The extraction of samples with methanol 80% and the reaction with the DPPH• solution was carried out as described by Pasqualone et al. (2014) [9], then the absorbance at 515 nm was read using a Cary 60 spectrophotometer (Agilent Technologies, Santa Clara, CA, USA). The results were expressed as µmol equivalents of 6-hydroxy-2,5,7,8-tetramethylchromane-2-carboxylic acid (Trolox, Sigma-Aldrich, Saint Louis, MO, USA). The analysis was carried out in triplicate.

### 2.5. Color Determinations

The colors of the samples were measured using a colorimeter (CM-600d, Konica Minolta, Tokyo, Japan) equipped with SpectraMagic NX software (Konica Minolta, Tokyo, Japan). The color coordinates *L** (lightness, from black to white), *a** (from green to red), and *b** (from blue to yellow) were determined in the CIE (Commission Internationale de l’Éclairage) color space. Seven replications were carried out on the flatbreads and four replications on the flours.

### 2.6. Image Analysis of Breads

The bread image analysis was carried out according to De Angelis et al. (2020) [12]. Since *laxoox* is typically pockmarked with holes, or “eyes” [3], and appears translucent when held up to a light source, the samples were placed on a white light LED transilluminator (Thermo Fisher Scientific, Osterode am Harz, Germany) to enhance the visibility of the eyes, and their images were acquired by a Sony α-6100 mirrorless camera (Sony Corporation, Tokyo, Japan) equipped with a Sony 16–50 mm f/3.5–5–6 lens (Sony Corporation, Tokyo, Japan). The images were processed using the ImageJ software (National Institutes of Health, Bethesda, USA). First, they were converted into an 8-bit grayscale. Then, an image section of 55 × 30 mm was cropped from the central area of each bread, filtered using the thresholding function to obtain the best cell resolution, and analyzed. The cells were grouped into five classes, i.e., 0.05 ≤ x < 0.5 mm^2^; 0.5 ≤ x < 1 mm^2^; 1 ≤ x < 5 mm^2^; 5 ≤ x < 10 mm^2^, and their percentages were determined along with the cell density, i.e., the ratio between the number of cells and the area used for the image evaluation. Three replicated analyses were carried out.

### 2.7. Statistical Analysis

The results were expressed as the mean ± standard deviation (SD). Analysis of variance (ANOVA) and Tukey’s HSD test were carried out by Minitab 19 Statistical Software (Minitab Inc., State College, PA, USA) and the differences were considered statistically significant at *p ≤* 0.05. The principal component analysis (PCA) was carried out to explore the dataset. In particular, the PCA was computed using Minitab 19 Statistical Software (Minitab Inc., State College, PA, USA), considering all the data except for the “thickness in spiral (mm)” and “total anthocyanins”, because those parameters were not available for all the samples. Given their different scales, the data were autoscaled before PCA.

## 3. Results and Discussion

### 3.1. Grain Diversity in the Starting Mix, and Other Ingredients

Table 1 reports the grain composition of the starting flour mix and details the other ingredients used in the preparation of the examined breads.

Somali *laxoox* flatbread is made from various, often low-gluten or no-gluten grains (cereals and legumes), mixed and milled in the same marketplace where they are bought. There is some variability in the mix according to local taste, family preference, and availability. Red and white sorghum, millet (pearl millet and finger millet), barley, corn, wheat, and pulses such as cowpea were recorded among the grains composing the flour mixture, locally named *budo*, confirming previous findings [3].

With the exception of wheat and corn, which are mostly imported and in the past were not used in *laxoox* preparation in Somaliland [3], the others are indigenous African grains, generally well adapted to drought, heat, and poor soils [13,14]. Currently, the *budo* flour blend is supplemented with refined wheat flour prior to breadmaking. Wheat provides gluten and imparts elasticity to achieve a soft and spongy bread texture. When refined wheat flour is available and affordable, its addition to the *laxoox* batter is preferred, although bread can also be made without it [3].

The *budo* flour blend does not contain salt, which is added at the time of breadmaking along with water. Some bread samples (F, G, H, and I), however, did not include salt. Water is used generally in an amount of two parts per each part of dry ingredients to obtain a fluid batter. Other ingredients are also eventually added to the batter, such as fenugreek, peanuts, black cumin, and black pepper, according to household preference. Of these, fenugreek and black pepper, mostly used as flavorings due to their intense aromas, have rich phytochemical profiles which makes them suitable not only for culinary, but also for medicinal purposes. Fenugreek seeds show antidiabetic and cholesterol-lowering, galactagogue, and carminative properties [15]. Black pepper has anti-inflammatory, antimicrobial, and antioxidant effects, and improves gastrointestinal health [16].

Yemeni *lahoh* flatbread C had a grain composition similar to that of Somali breads, but with the addition of millet, while Yemeni bread B was the only one that contained an ingredient of animal origin, i.e., cow yogurt. Bread sample B also contained black cumin seeds, which have health properties similar to those of black pepper.

### 3.2. Nutritional Characteristics and Bioactive Compounds

All the bread samples, and, for those coded E, F, G, and I, also the starting flours, were submitted to the analysis of nutritional composition. The sampling plan included breads (27 samples in total, of which 21 were Somali *laxoox* and 6 were Yemeni *lahoh*) processed by different households, with slight differences in the processing methods and with different raw materials, to represent the variability of bread characteristics among different producers and to give an insight into the culinary traditions and cultures of Somalia and Yemen.

One sample of commercial *budo* flour blend (sample A) was also analyzed. The latter was solely composed of red sorghum and, prior to bread-making, was mixed with wheat flour.

Table 2 reports the nutritional composition of the flours and breads. The protein content of the collected flours (9.41–15.65 g proteins/100 g) reflected their compositional grains, which were sorghum (having a protein content approximately ranging from 4 to 11 g proteins/100 g) [13]; corn (7–10.5 g/100 g) [17]; barley (8.5–14.5 g proteins/100 g) [18]; and wheat (8–14 g proteins/100 g) [19].

The flour sample G showed the highest protein content (15.65 g proteins/100 g) due to the presence of cowpea, with an average content of 24 g proteins/100 g [14].

The protein content of the bread samples was between 12.47 and 15.94 g proteins/100 g d.m., with significant differences (*p* ≤ 0.05) among households, imputable to their unique formulations. Among the Somali breads, the highest value (>15 g proteins/100 g) was found in the bread samples F and G, without a significant difference between them. These breads were either prepared by using protein-rich additional ingredients (sample F) or by using flour mixes with high protein content (sample G).

A higher protein content was observed in bread compared to the starting flour (when available), especially for samples A, E, and F. This increase can be explained with the addition, prior to bread-making, of other ingredients with a higher protein content than the flour mix, such as ground peanuts and fenugreek (sample F), or by the transfer of protein material from artisanal, imperfectly clarified ghee (still containing residual proteins), commonly used to grease the baking griddle (sample E). Peanuts and fenugreek, indeed, have average protein contents in the range of 24–29 g proteins/100 g and 27–32 g proteins/100 g, respectively [15,20]. In the case of bread sample A, the increase in protein content compared with the starting *budo* blend, composed only of red sorghum, was due to the addition of wheat flour, which generally has a higher protein content than sorghum.

The Yemeni breads, whose starting flours were unavailable at the time of collection, showed a high protein content (around 15 g proteins/100 g), likely imputable to the inclusion of cow yogurt in the batter (sample B) or, again, due to contaminations from ghee (sample C).

The lipid content of the flours ranged from 2.15 to 2.90 g lipids/100 g, and, similarly to proteins, reflected the fat content of raw materials. Indeed, sorghum contains 1–3 g lipids/100 g [13], about 2 g lipids/100 g cowpea [14], 2–4 g lipids/100 g corn [21], and 2–3 g lipids/100 g barley and wheat [22].

The lipid content of the breads was slightly higher than that of the corresponding flours, due to the transfer of some fat from the greased baking griddle. Bread samples D and F showed a significantly higher lipid content than the other breads. The lipid content of bread sample F was particularly high (4.11 g lipids/100 g), probably linked to the addition of ground peanuts, while for the sample D, which did not include fatty ingredients, the transfer of fat from the baking plate could have been more relevant. The Yemeni samples showed a lipid content in the same range of the Somali breads, with sample B being richer in fats, probably due to the lipids contributed by the cow yogurt.

The fiber content of the flours ranged from 5.14 to 8.61 g fibers/100 g, while that of the bread samples ranged from 4.72 to 7.03 g fibers/100 g. For bread, the highest fiber content was observed in sample I, which was prepared without refined wheat flour. In the other bread samples, instead, the presence of additional ingredients slightly lowered the fiber content compared to the collected flour. Flour sample A, whose fiber content agreed with the mean value reported for sorghum [23], was mixed with refined wheat flour prior to breadmaking, so the fiber content of bread decreased. Flour sample F, being mixed with black pepper, fenugreek, and peanuts, showed very high fiber content, but the fibers decreased in the final bread and fat increased, which was likely due to a relevant transfer from the greased baking plate. Dietary fiber reduces the glycemic index of foods, keeps the gut healthy, and lowers the blood cholesterol levels [23,24]. Fiber fortification of foods is, therefore, a current trend of the food industry in Western countries to prevent cardiometabolic diseases [25,26,27]. In the Horn of Africa, however, more energetic foods are generally required to prevent malnutrition. In Somaliland, indeed, and even more in south-central Somalia, recurrent droughts and flooding affect food security [28] and malnutrition is a serious public health problem, especially for children [29].

The examined flatbreads showed slightly higher contents of proteins and lipids, and slightly lower fiber, than the mean values reported for wholemeal wheat-based bread in the Italian database of food composition [30]. However, beyond than a mere nutritional comparison, other aspects must be taken into account. The most important feature of these breads, in fact, was that they were free from any additives (in many cases, even salt), and that they were prepared with a liquid microbial starter with acidifying properties, similar to sourdough.

Food systems around the world have industrialized over the last century, with impacts on diet, nutrition, and health. One result is the commercial production of bread made from refined wheat flour, produced with compressed yeast [31] often high in salt [32] and containing additives, such as calcium propionate or potassium sorbate, and mono- and diglycerides of fatty acids [33]. The combination of sourdough-based breadmaking with unrefined flour, in contrast, is known to be a healthier option [34]. Therefore, maintaining the original nature of the Somali flatbread could have positive effects on health.

Besides nutrients, foods may contain several radical scavenging compounds, such as phenolics, carotenoids, and anthocyanins. The examined samples contained interesting amounts of bioactives, higher in the flours than in the corresponding breads. Due to the known association between oxidative stress and cell damage and subsequent pathogenesis of various disorders and diseases [35], the content of antioxidants in a food is as important as its content of nutrients.

As for carotenoids (Table 3), flour sample A (commercial *budo*) showed the highest content of these compounds (22.58 mg β-carotene/kg) among all the examined flours. This sample was actually composed of pure red sorghum flour, known to be rich in carotenoids—mostly zeaxanthin, followed by β-carotene and lutein—which are responsible for the typical yellow–orange coloring of the grain. Also, the other flour samples contained carotenoid-rich grains, namely, red sorghum or yellow corn, but in mixture with refined flour, which lowered the carotenoids to 4.10–9.34 mg β-carotene/kg due to a dilution effect.

The carotenoid content of the Somali breads ranged from 2.44 to 10.48 mg β-carotene/kg, with the highest value in bread sample A, followed by sample I. Bread sample A, however, showed half of the carotenoid content of the corresponding flour, composed of pure red sorghum flour, because prior to bread-making, the latter was mixed with refined wheat flour (which does not contain carotenoids) according to the “modern style” for Somali *laxoox* breadmaking [3]. In addition to this dilution effect, the prolonged exposure of the batter to air during mixing and leavening, as well as the typical large and thin-layered shape of flatbread, which further exposes a large surface area to air during baking, was probably the cause of the extensive oxidative degradation of carotenoids. This kind of degradation could be both lipoxygenase-mediated and non-enzymatic [36], not to mention that heat treatment also has a detrimental effect on carotenoids [37]. On the contrary, bread sample F did not show a marked decrease in carotenoids compared to the flour, likely due to the addition of fenugreek seeds, the typical yellow color of which is due to β-carotene [15]. Bread sample I showed the second highest carotenoid content among Somali breads, because it was not “diluted” with refined wheat flour.

Regarding the two Yemeni breads, the carotenoid content was significantly different between them, with sample C in the range of the Somali ones, while sample B showed a very high content, the highest among all the examined breads. The sources of carotenoids in Yemeni bread sample B could have been corn or cow yogurt, both used in its preparation.

As for the phenolic compounds, their levels in the flours were in the range 6.82–7.42 mg GAE/g, without significant differences among the samples. All cereal grains contain phenolic compounds, mainly located in the husk and the aleurone layer [38]. Sorghum, in particular, is known to be a rich source of phenolic compounds, although the content varies significantly in different varieties (4.33–89.2 mg GAE/g) [39]. The phenolics in wheat, instead, are in the range of 1.28–3.15 mg GAE/g [9].

The levels of phenolics in the Somali breads were lower than in the flours, in the range of 5.02–6.15 mg GAE/g. The phenolic compounds are mostly found in so-called “bound” forms, linked to the arabinoxylans of the cell walls [38], and could be partially released by the effect of sourdough fermentation [40], which was the type of fermentation carried out in the examined breads, with the exception of bread samples C and I. However, although the release of bound phenolics should have increased their contents in the breads, compared to the flours, the combined effect of air exposure and thermal degradation caused a decrease in phenolics during baking, which is in agreement with previous studies [41].

The content of phenolics in the Yemeni breads paralleled that of carotenoids, with sample C having a content similar to the Somali breads, while sample B had a significantly higher content, probably derived from the black cumin seeds added prior to bread-making.

The anthocyanins were found in negligible amounts in the flours, with the exception of the *budo* flour blend sample A. Therefore, anthocyanins were not determined in the breads, as they were likely absent due to the detrimental effect of heat on these compounds [42]. The total anthocyanins of flour sample A (pure red sorghum) were the most concentrated bioactive compounds, accounting for 0.3 mg cyanidin 3-*O*-glucoside/g on dry matter. In agreement with our findings, amounts of anthocyanins of 0.1–0.7 mg/g were reported by other authors in red sorghum, with higher amounts in the brown and black varieties [43]. The most abundant anthocyanin compounds of sorghum are the 3-deoxyanthocyanidins and their derivatives, such as luteolinidin (orange) and apigeninidin (yellow) [38], which are not commonly found in higher plants. Anthocyanins impart a typical red color to foods and beverages, which becomes blue when the pH becomes basic. 3-Deoxyanthocyanidins, instead, are more stable to pH changes, and therefore have better potential as natural food pigments [43]. Refined wheat flour does not contain anthocyanin compounds unless obtained by milling pigmented (red, purple, or blue) varieties.

Antioxidants interrupt the radical reaction by donating hydrogen atoms or electrons to the radicals and leading to the formation of more stable compounds. The values of in vitro antioxidant activity measured by the DPPH assay, which is based on the ability of food bioactives to scavenge the DPPH radical, were generally higher in the flours than in the breads. The flour samples showed antioxidant activity in the range of 0.71–2.38 µmol TE/g. The flours A and I showed the highest antioxidant activity, without significant differences between them, because they were not diluted with refined wheat flour. The antioxidant activity resulted from the combination of the bioactives contributed by the different grains and the other ingredients. Values of antioxidant activity between 1.32 and 4.68 µmol TE/g have been determined in red sorghum [44], 1.3–1.6 µmol TE/g in barley [45], and 1.48–2.28 µmol TE/g in whole wheat [9], but this value lowers to 0.86 µmol TE/g in refined wheat flour [12].

The antioxidant activity of the Somali breads was in the range of 0.36–1.53 µmol TE/g, with the highest value in sample I, which was prepared without adding refined wheat flour. As for the Yemeni breads, sample C had an antioxidant activity in the same range of the Somali breads, while bread sample B had significantly higher antioxidant activity (2.84 µmol TE/g), related to its content of carotenoids and phenolics, which were the highest among the examined samples. Moreover, the antioxidant activity of this sample could come from the bioactive peptides of yogurt, released during lactic fermentation [46]. The observed values of antioxidant activity were comparable to those ascertained in breads prepared by partially (15%) replacing water with vegetable juices (e.g., tomato, carrot) [47].

In addition to the initial antioxidants of flour, the Maillard reaction products which originate during baking, such as sugar reductions and melanoidins, can contribute to the antioxidant capacity of bread [48]. With the exception of bread sample I, which was dark brown (Figure 1), the examined breads were generally pale brown, indicating that the Maillard reaction did not occur to a high degree. Sample I, indeed, showed a higher antioxidant activity which, therefore, could have been partly due to the Maillard reaction products.

### 3.3. Physical Characteristics Related to the Appearance of Breads

The main physical characteristics pertaining to the appearance of the bread samples are reported in Table 4.

Once fermented, the *laxoox* batter is poured onto the center of a flat cast iron griddle called a *dawa* (also spelled *dhaawe*), prepared with ghee, and is spread using a spiral motion to make a thin, single-layered flatbread with a typical spiral relief pattern [3]. The presence of a thicker, sometimes slightly darker, spiral pattern on the surface was observed in all the sampled Somali breads. This feature is considered an important quality trait of Somali *laxoox* [3]. The bread thickness ranged from 3.3 to 5.1 mm in the spiral (Table 1), which was the thickest part. The diameter varied from 13.4 to 20.2 cm, with a slightly greater variability than the 15–19 cm range recorded in a previous survey [3].

As for the Yemeni bread samples, the *Sana’ani* style *lahoh* (sample C) had a diameter of 15 cm and a thickness of 4 mm, similar to the Somali *laxoox* samples, but did not show the spiral pattern (Figure 1). Therefore, if the nutritional characteristics and the content of bioactive compounds were a result of each unique household recipe, the physical characteristics also depended on the manual input of the households.

The *lahoh Ariiqy* (sample B) was very different in size from the other bread samples. This slightly oval bread, with a length of 55 cm, was the largest sample collected. This size is quite common for breads baked on the *sajj*, a 70 cm large circular and convex metal griddle [49]. The breadth of *lahoh Ariiqy* was accompanied by a remarkable and uniform leanness (1.1 mm). Furthermore, the surface of the *lahoh Ariiqy* was characterized by a unique appearance, with dark parallel stripes featuring a typical “zebra” pattern (Figure 1). This bread is prepared from a sticky dough, and the way in which this dough is spread on the *sajj*, with a back-and-forth motion, determines the formation of these stripes.

The image analysis showed that the *laxoox* bread crumb was characterized by variable porosity in terms of cell density and cell size distribution, although the majority of the cells were very small, ranging from 0.05 to 0.5 mm^2^ (Table 4). The cell density of the *laxoox* sample G (57.4 cell/cm^2^) was significantly higher (*p ≤* 0.05) than that of samples A and F (22.4 and 31.9 cell/cm^2^, respectively). Therefore, G bread had a closer and finer porosity than A and F, which, instead, showed more open grains with coarser and less numerous pores (Figure 2). The other samples (D, E, H, and I) had intermediate values of cell density and did not show significant differences among them.

The cells with sizes of 0.05–0.5 mm^2^ ranged from 75.9 to 91.1%, with the highest value in the sample H, without significant differences with D. The 0.5–1 mm^2^ cells and the 1–5 mm^2^ cells were found in similar amounts (about 4–14.5%), and negligible quantities of cells sized 5–10 mm^2^ were observed. These findings show that all the examined breads, although with some differences, had porous structures. Indeed, the CO_2_ released during the prolonged fermentation, and the air incorporated during the vigorous manual mixing of the batter, were trapped by the matrix composed of gluten (from the refined wheat flour added and from the wheat and barley grains eventually included in the *budo* blend) and pre-gelatinized starch (from a thermally treated portion of dough which is traditionally added to the batter, known as *cajiin*) [3]. They then thermally expanded during baking, forming visible pores. Abundant porosity, with the presence of numerous cells called “eyes”, leading to a spongy structure, is another important quality feature of *laxoox* flatbread, together with the spiral pattern [3].

The Yemeni *lahoh* flatbreads were also characterized by a porous structure. The bread from Sana’a (sample C) showed an open crumb structure, having the highest number of the largest cells (1–5 mm^2^ and 0.5–1 mm^2^), and the lowest number of the smallest cells (0.05–0.5 mm^2^) among all the bread samples analyzed. These findings were probably imputable to the use of commercial baker’s yeast in the preparation of this bread sample, causing a more pronounced fermentation, with larger pores. However, the differences between the Yemeni bread C and the Somali ones (generally fermented with a microbial starter from the previous batch, called *dhanaanis*, without commercial yeast) were not always statistically different, indicating that *Sana’ani*-style Yemeni *lahoh* shares some similar features with Somali *laxoox*, the latter being a kind of bread which has its own variability. The cell density of the Yemeni sample C, indeed, was statistically similar to that of the Somali breads, except sample G. On the contrary, Yemeni sample C was significantly different from Yemeni sample B across all image analysis data, namely, cell density and cell size distribution. The Yemeni sample B (*lahoh Ariiqy*), indeed, showed a very high cell density, accounting for 145 cell/cm^2^, which was the highest value among all the samples examined. Its cells were very small (93.7% were in the range of 0.05–0.5 mm^2^, with no cells larger than 5 mm^2^); therefore, this bread had the finest pore grains (Figure 3).

This bread contained cow yogurt, contributing both triglycerides and phospholipids, the latter being responsible for emulsifying and stabilizing the incorporation of air deriving from beating. Carbon dioxide arose from fermentation. A decrease in the size of the cells has been reported by other authors as a consequence of the addition of fats to bread dough [50].

As for color (Table 5), the flours were paler (higher values of *L** and lower values of *a** and *b**) than the corresponding breads due to enzymatic browning effected by polyphenol oxidase during batter preparation and fermentation, as well as non-enzymatic browning related to the Maillard reaction during baking.

The A flour sample, which was constituted by red sorghum, without the addition of wheat flour, showed the highest value of *a**, indicating a reddish hue (Figure 4). Indeed, this sample showed higher values of anthocyanins than the other flours (Table 3). This sample also showed a very high carotenoid content, but its *b** value, indicating the yellow hue, was not accordingly high, probably because the red hue masked the yellow one.

As for the Somali breads, significant differences were observed among the samples due to the different compositions of the starting flour and variations in the baking process. The bread sample I was the darkest (*L** = 32.92), while D, H, and E were the lightest in color (*L** = 50.88, 50.71, and 48.39, respectively). The dark brown color of the sample I was due to the absence of refined wheat flour in its formulation, while E and H contained the highest proportion of refined wheat flour (which was not specified in D). The majority of the breads showed more intensely colored (lower *L** and higher *a** and *b** values) bottom sides than top sides, while the bottoms were paler in samples G and H. The thin, pressed spiral of the *laxoox* samples was generally slightly darker than the rest of the surface, with negligible differences for samples E and H. As for the *a** and *b** color coordinates, they always showed positive values, with significantly higher values in sample D than in the other Somali breads, indicating more intense red and yellow hues, respectively.

The *lahoh* bread from Sana’a (bread sample C) did not show significant differences in the values of *L**, *a**, or *b** compared to the Somali breads. The *lahoh Ariiqy* (sample B) showed a stripe pattern much darker than the rest of the surface, which, in contrast, was bright yellowish, with the highest value of *L** and *b** among all the samples examined, probably due to the presence of corn in the starting grain mixture.

### 3.4. Comparison of the Breads

The *lahoh* bread samples, originally from Yemen, have a similar name to Somali *laxoox*. They were collected in order to compare them with the Somali bread samples. The entire dataset composed of the chemical and physical results was subjected to principal component analysis (PCA), which helped in identifying the similarities and differences among samples.

Figure 5A,B shows the observation scores for the first two components, accounting for 58.45% of variability.

In particular, PC1 enabled the highlighting of the main differences between the Somali *laxoox* and the Yemeni *lahoh* breads, mostly with regard to diameter, cell density, and antioxidant activity. The Yemeni *lahoh Ariiqy* bread (sample B) was particularly distant in characteristics from all the Somali *laxoox* ones, while the *Sana’ani* style Yemeni *lahoh* bread (sample C) was closer to them. The latter is an ordinary *lahoh*, commonly prepared everywhere in Yemen. The *lahoh Ariiqy*, instead, is a highly unique version of *lahoh* prepared only in Aruuq Village. The household providing *lahoh Ariiqy* described it as a “special” bread that would not easily be found anywhere else. Accordingly, PCA demonstrated this uniqueness.

Interestingly, the Somali *laxoox* group presented a certain variability, with samples G and H being related to each other and forming a separate subgroup. This subgroup was distinguishable along the PC2 from the majority of the other Somali samples, and it was influenced by higher values of the color indices *L** and *b** of the bottom side, as well as, secondarily, the cell size class 0.05–0.5 mm^2^, and on the other hand, by the lower values of total carotenoids, total phenolics, and antioxidant activity. Some variability is expected in food products with an artisanal character, with a strong manual input.

The breadmaking process in Yemen and Somaliland is largely similar for *laxoox/lahoh*, including the selection of ingredients, fermentation steps, and baking, as well as the ways in which household technologies domesticated in Yemen and Somaliland in the last century (e.g., blenders, refrigeration) have given rise to new techniques. These similarities contrast with the production of the same bread in Somalia, which goes by the name “*canjeero*” and involves different dry ingredients and fermentation steps [3]. In the fermentation stage, while the use of pre-gelatinized dough in the initial sourdough batch fell out of favor circa the 1980s in Somaliland [3], it continues to be used in Yemen. In both locations, long fermentation and a microbial starter are used in subsequent batches. Other differences are readily identifiable, especially at the baking stage. Somaliland baking tools (e.g., cast iron pans of a certain diameter, flat-bottomed plastic cups to pour and spread the batter across the pan, and butter knives to lift the cooked bread from the pan) are consistent across households. In Yemen, by contrast, the baking pan size differs drastically depending on the number of people to serve and on commercial versus household production. A special tool is often used to drip batter onto the pan, where it is spread by manipulation (swirling) of the pan itself. In the finished product, apart from differences in size, a spiral pattern is characteristic of Somaliland *laxoox*, whereas this pattern appears inconsistently in Yemeni *lahoh*. Other patterns (i.e., the “zebra stripe” featured in this article) indicate highly localized variations (i.e., *lahoh* Ariiqy).

## 4. Conclusions

Our results show that the nutritional characteristics of breads, the bioactive content, and the antioxidant activity significantly varied among households and baking batches due to the natural variability of manual food preparation, as well as in the recipes and compositions of raw materials used. Interestingly, the total carotenoids were the highest (22.58 mg β-carotene/kg) in red sorghum flour, where anthocyanins were also found (0.32 mg cyanidin 3-*O*-glucoside/g), but markedly decreased by adding refined wheat flour, indicating that this modern habit has negative effects on the content of bioactive compounds.

All of the breads had porous structures, with a cell density varying from 22.4 to 57.4 cells/cm^2^ in the Somali *laxoox*, while one of the two Yemeni *lahoh* reached 145 cells/cm^2^.

The PCA highlighted the variability among the *laxoox* Somali flatbreads, with two samples forming a separate subgroup. Moreover, the PCA underlined the main features that distinguish the *laxoox* breads from both of the Yemeni *lahoh* samples, although one of the latter, representative of the ordinary flatbread production in Yemen (the *Sanaa’ani* style *lahoh*), was not very distant from the *laxoox* main group.

Somali and Yemeni flatbreads remain largely unexplored to date. The results of this investigation have both technical and cultural implications by shedding light on the physical–chemical and nutritional properties of these breads and by offering valuable insights into their social and ethnographic significance. These flatbreads are still very artisanal and not at all standardized, so at this stage, it was important to assess the extent of the variability of their characteristics. “Characterization” studies are quite common in the scientific literature as preparatory studies prior to further deepening. Therefore, the improvement of the nutritional quality of these breads could be the subject of further studies.

Finally, though scarce in number due to security challenges which hindered sample collection in the area of origin, the Yemeni samples provide points of contrast with the Somali flatbread, which deserve further investigation that could be integrated with ethnographic and historical studies.

## Figures and Tables

**Figure 1 foods-12-03050-f001:**
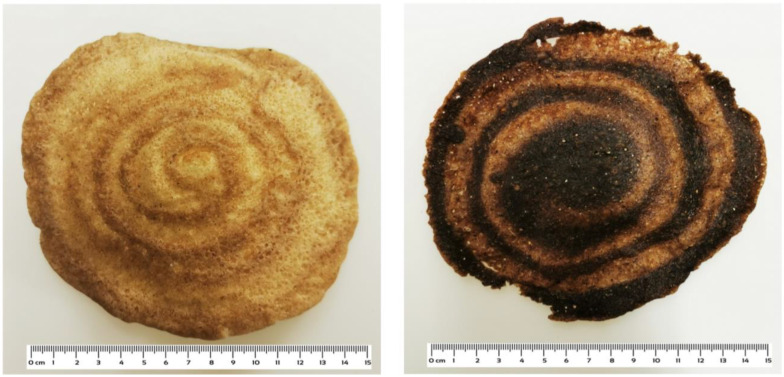
Somali *laxoox* bread sample G (**left**) and Somali *laxoox* bread sample I (**right**).

**Figure 2 foods-12-03050-f002:**
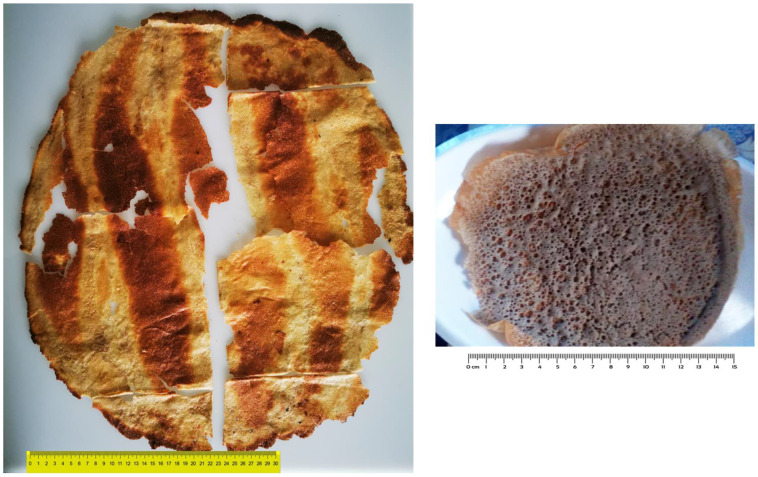
(**Left**): Yemeni *lahoh Ariiqi* bread, meaning “*lahoh* bread from Aruuq village” (near Tais, Yemen). This bread is very thin, and immediately after baking, when still hot and slightly flexible, it is folded like a handkerchief for the sake of storage or travel. The bread sample in the picture has been unfolded to show its typical “zebra pattern”. Some parts are missing due to breaking during unfolding. (**Right**): Yemeni *lahoh* bread *Sana’ani* style.

**Figure 3 foods-12-03050-f003:**
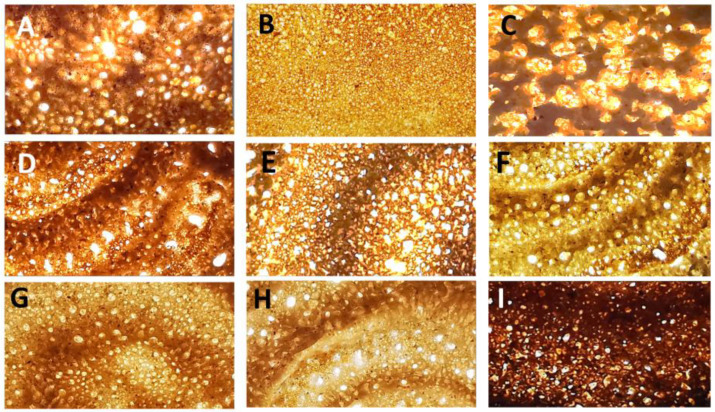
Image of the sections (55 × 30 mm) of Somali and Yemeni breads, viewed over a white light transilluminator to enhance the visibility of the pores. (**A**,**D**–**I**) = samples of Somali *laxoox* bread; (**B**,**C**) = samples of Yemeni *lahoh* bread.

**Figure 4 foods-12-03050-f004:**
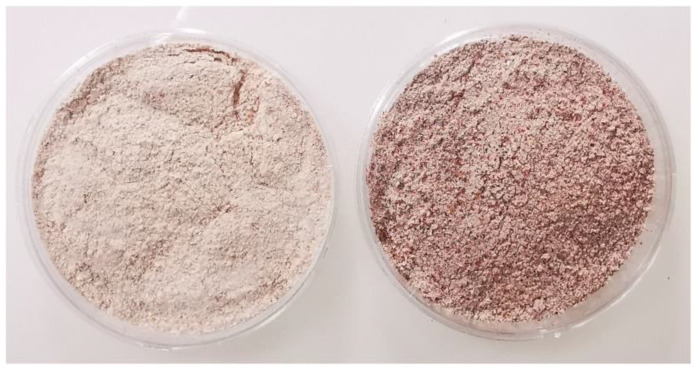
(**Left**): Flour sample F, used in the production of *laxoox* Somali bread, was constituted by white and red sorghum, barley, and wheat grains milled together, mixed with refined wheat flour. (**Right**): Flour sample A, used in the production of *laxoox* Somali bread was constituted by red sorghum without the addition of refined wheat flour.

**Figure 5 foods-12-03050-f005:**
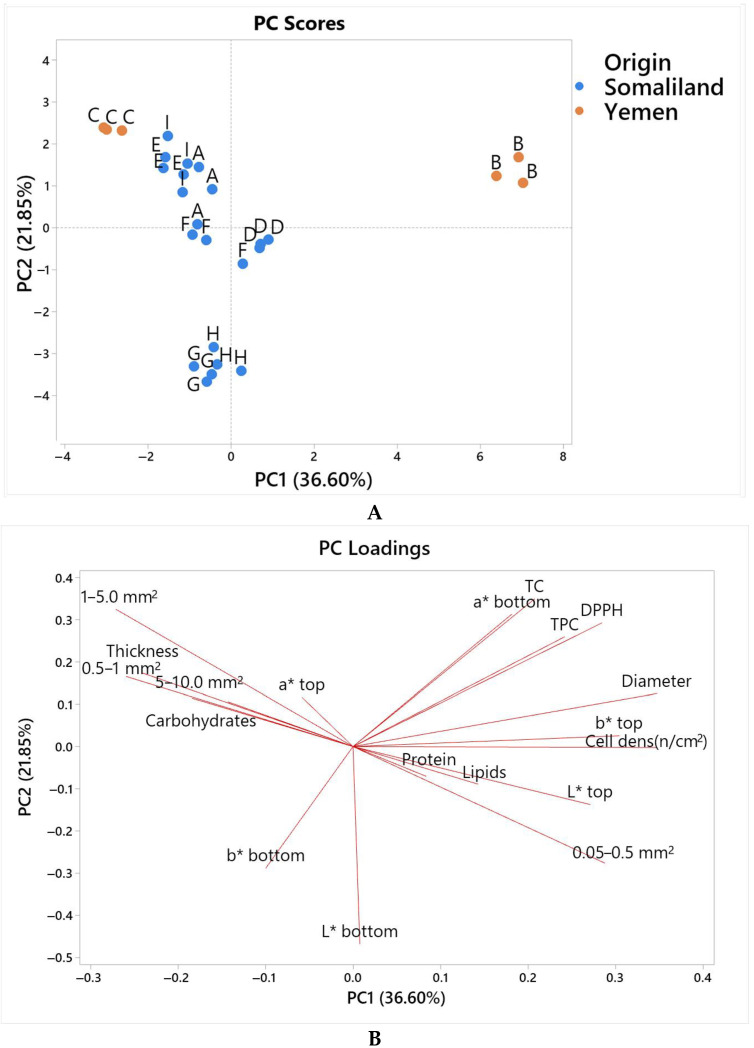
(**A**) Scatter plot for the first two principal components, PC1 and PC2, obtained for the entire dataset of physical–chemical and nutritional characteristics of Somali and Yemeni breads. A, D, E, F, G, H, and I = samples of Somali *laxoox* bread; B and C = samples of Yemeni *lahoh* bread. (**B**) Variable loading plot for the first two principal components, PC1 and PC2, obtained for the same dataset. *L**, *a**, *b** = color coordinates; DPPH = 2,2-diphenyl-1-picrylhydrazyl radical scavenging test; TC = total carotenoids; TPC = total phenolic compounds.

**Table 1 foods-12-03050-t001:** Composition of *budo* blends (traditional mixture of different grains milled together) and amount of refined wheat flour and other ingredients used to prepare the samples of Somali *laxoox* flatbreads (A, D, E, F, G, H and I) and Yemeni *lahoh* flatbreads (B and C).

Code	Grain Composition of the *Budo* Flour Mixture	Ratio of Refined Wheat Flour: *Budo* Flour	Other Ingredients
A	Red sorghum	1:1	Microbial starter from previous batch, salt
B	Wheat and corn	Not specified	Cow yogurt, black cumin seeds, microbial starter from previous batch
C	White and red sorghum, millet	1:1	Commercial baker’s yeast (*Saccharomyces cerevisiae*), salt
D	Red sorghum	Not specified	Salt, microbial starter from previous batch
E	Red sorghum	2:1	Salt, microbial starter from previous batch
F	White and red sorghum, barley and wheat	1:1	Ground peanuts, ground fenugreek, black pepper, microbial starter from previous batch
G	Yellow corn, cowpea, wheat	1:1	Microbial starter from previous batch
H	Barley	3:1	Microbial starter from previous batch
I	Red sorghum, white corn, barley and wheat	None	Overnight fermentation without starter

**Table 2 foods-12-03050-t002:** Nutritional composition of flours and breads (g/100 g dry matter).

Sample	Proteins	Lipids	Fibers	Available Carbohydrates
Flours				
A *	9.41 ± 0.38 ^d^	2.59 ± 0.26 ^abc^	6.68 ± 0.47 ^b^	88.00 ± 0.64 ^a^
E	11.50 ± 0.69 ^c^	2.21 ± 0.14 ^bc^	5.14 ± 0.23 ^c^	86.29 ± 0.83 ^b^
F	13.06 ± 0.41 ^b^	2.90 ± 0.27 ^a^	8.61 ± 0.46 ^a^	84.04 ± 0.14 ^c^
G	15.65 ± 0.18 ^a^	2.15 ± 0.17 ^c^	6.06 ± 0.51 ^bc^	82.19 ± 0.01 ^d^
I	12.58 ± 0.43 ^bc^	2.76 ± 0.25 ^ab^	6.76 ± 0.84 ^b^	84.65 ± 0.18 ^c^
Breads **				
A	12.97 ± 0.43 ^b^	2.67 ± 0.02 ^c^	4.72 ± 0.17 ^d^	84.36 ± 0.45 ^ab^
B	15.06 ± 0.83 ^a^	3.14 ± 0.08 ^b^	6.13 ± 0.25 ^b^	81.81 ± 0.74 ^c^
C	14.72 ± 0.47 ^a^	2.52 ± 0.21 ^cd^	5.49 ± 0.13 ^c^	83.45 ± 0.61 ^ab^
D	12.78 ± 0.33 ^b^	3.89 ± 0.03 ^a^	6.29 ± 0.40 ^ab^	83.33 ± 0.30 ^b^
E	12.99 ± 0.28 ^b^	2.61 ± 0.27 ^cd^	5.28 ± 0.13 ^c^	84.40 ± 0.01 ^ab^
F	15.25 ± 0.02 ^a^	4.11 ± 0.01 ^a^	6.85 ± 0.15 ^a^	80.63 ± 0.03 ^c^
G	15.94 ± 0.34 ^a^	2.47 ± 0.05 ^d^	6.02 ± 0.12 ^b^	81.59 ± 0.29 ^c^
H	12.47 ± 0.55 ^b^	2.86 ± 0.05 ^bcd^	4.75 ± 0.33 ^d^	84.68 ± 0.50 ^a^
I	12.70 ± 0.11 ^b^	2.89 ± 0.01 ^bc^	7.03 ± 0.34 ^a^	84.41 ± 0.39 ^ab^
Range Somali (min–max)	12.47–15.94	2.47–4.11	4.72–7.03	80.63–84.68

* Sample of commercial *budo* composed only of red sorghum without the addition of refined wheat flour; ** A, D, E, F, G, H, and I = Somali *laxoox* bread; B and C = Yemeni *lahoh* bread. Data are presented as means ± standard deviation of three samples. Different letters in a column for the same group of data indicate statistical differences at *p* ≤ 0.05.

**Table 3 foods-12-03050-t003:** Bioactive compounds and antioxidant activity of flours and breads.

Sample	Total Carotenoids (mg β-Carotene/kg d.m.)	Total Phenolic Compounds (mg GAE/g d.m.)	Total Anthocyanins (mg Cyanidin 3-*O*-Glucoside/g d.m.)	Antioxidant Activity * (µmol TE/g d.m.)
Flours				
A **	22.58 ± 0.11 ^a^	7.38 ± 0.24 ^a^	0.32 ± 0.01 ^a^	2.15 ± 0.01 ^a^
E	6.61 ± 0.11 ^d^	6.93 ± 0.07 ^a^	0.01 ± 0.00 ^b^	0.72 ± 0.07 ^c^
F	7.88 ± 0.17 ^c^	6.82 ± 0.23 ^a^	0.02 ± 0.01 ^b^	1.26 ± 0.03 ^b^
G	4.10 ± 0.01 ^e^	7.42 ± 0.06 ^a^	n.d.	0.71 ± 0.04 ^c^
I	9.34 ± 0.16 ^b^	7.21 ± 0.04 ^a^	0.03 ± 0.01 ^b^	2.38 ± 0.10 ^a^
Breads ***				
A	10.48 ± 0.24 ^b^	5.55 ± 0.66 ^bc^	n.d.	1.06 ± 0.07 ^c^
B	12.51 ± 0.17 ^a^	7.11 ± 0.12 ^a^	n.d.	2.84 ± 0.09 ^a^
C	5.86 ± 0.01 ^f^	6.01 ± 0.11 ^b^	n.d.	1.14 ± 0.03 ^c^
D	6.77 ± 0.18 ^e^	6.07 ± 0.03 ^b^	n.d.	1.08 ± 0.05 ^c^
E	6.18 ± 0.17 ^e^	5.85 ± 0.12 ^b^	n.d.	0.76 ± 0.10 ^d^
F	7.44 ± 0.25 ^d^	6.15 ± 0.08 ^b^	n.d.	0.82 ± 0.04 ^d^
G	2.49 ± 0.05 ^g^	5.02 ± 0.15 ^c^	n.d.	0.36 ± 0.02 ^e^
H	2.44 ± 0.06 ^g^	5.56 ± 0.26 ^bc^	n.d.	0.42 ± 0.06 ^e^
I	8.37 ± 0.05 ^c^	5.69 ± 0.43 ^bc^	n.d.	1.53 ± 0.08 ^b^
Range Somali (min–max)	2.44–10.48	5.02–6.15	-	0.36–1.53

* Determined by the 2,2-diphenyl-1-picrylhydrazyl (DPPH) radical scavenging test. ** Sample of commercial *budo* composed only of red sorghum without the addition of refined wheat flour; *** A, D, E, F, G, H, and I = Somali *laxoox* bread; B and C = Yemeni *lahoh* bread; GAE = gallic acid equivalents; TE = Trolox equivalents (Trolox = 6-hydroxy-2,5,7,8-tetramethylchroman-2-carobxylic acid); n.d. = not determined. Data are presented as means ± standard deviation of three samples. Different letters in a column, for the same group of data, indicate statistical differences at *p* ≤ 0.05.

**Table 4 foods-12-03050-t004:** Main physical characteristics related to the appearance of bread samples. A, D, E, F, G, H, and I = Somali *laxoox* bread; B and C = Yemeni *lahoh* bread.

Code	Typical Surface Pattern	Thickness (mm)	Thickness in Spiral (mm)	Diameter (cm)	Cell Size Distribution (%)				Cell Density (Cells/cm^2^)
					0.05–0.5 mm^2^	0.5–1 mm^2^	1–5 mm^2^	5–10 mm^2^	
A	Spiral relief	2.1 ± 0.1 ^b^	5.1 ± 0.1 ^a^	16.3 ± 0.6 ^c^	76.2 ± 3.5 ^de^	9.5 ± 2.2 ^cd^	13.4 ± 1.5 ^ab^	0.4 ± 0.4 ^bc^	22.4 ± 7.7 ^c^
B	Parallel stripes (zebra effect)	1.0 ± 0.1 ^c^	n.d.	55.0 ± 1.1 ^a^*	93.7 ± 0.5 ^a^	4.5 ± 0.3 ^e^	1.8 ± 0.3 ^d^	0 ± 0 ^c^	145 ± 15 ^a^
C	None	4.0 ± 0.2 ^a^	n.d.	15.0 ± 0.1 ^d^	65.7 ± 1.5 ^f^	15.0 ± 1.2 ^a^	17.3 ± 0.1 ^a^	1.4 ± 0.3 ^a^	27.5 ± 1.5 ^c^
D	Spiral relief	2.0 ± 0.1 ^b^	4.0 ± 0.8 ^bc^	16.5 ± 0.4 ^c^	84.4 ± 0.7 ^bc^	7.6 ± 1.1 ^de^	6.3 ± 0.1 ^cd^	1.1 ± 0.1 ^ab^	44.1 ± 0.9 ^bc^
E	Spiral relief	2.1 ± 0.1 ^b^	4.8 ± 0.5 ^ab^	20.2 ± 0.4 ^b^	68.9 ± 1.9 ^ef^	14.1 ± 0.6 ^ab^	15.2 ± 1.1 ^a^	1.3 ± 0.1 ^a^	42.9 ± 4.7 ^bc^
F	Spiral relief	1.1 ± 0.1 ^c^	3.3 ± 0.5 ^c^	13.4 ± 0.3 ^e^	75.9 ± 3.3 ^e^	12.2 ± 2.6 ^abc^	9.9 ± 3.1 ^bc^	1.2 ± 0.6 ^ab^	31.9 ± 7.5 ^c^
G	Spiral relief	1.7 ± 0.6 ^b^	4.1 ± 0.1 ^abc^	14.5 ± 0.1 ^d^	83.4 ± 2.3 ^cd^	12.1 ± 1.2 ^abc^	4.3 ± 0.9 ^d^	0.1 ± 0.0 ^c^	57.4 ± 4.6 ^b^
H	Spiral relief	2.0 ± 0.1 ^b^	4.7 ± 0.6 ^ab^	17.1 ± 0.4 ^c^	91.1 ± 2.1 ^ab^	4.3 ± 0.6 ^e^	3.9 ± 1.6 ^d^	0.7 ± 0.1 ^abc^	49.2 ± 19.4 ^bc^
I	Spiral relief	2.1 ± 0.1 ^b^	4.1 ± 0.1 ^abc^	15.2 ± 0.2 ^d^	76.3 ± 4.6 ^de^	10.6 ± 1.6 ^bcd^	12.8 ± 2.8 ^ab^	0.2 ± 0.2 ^c^	33.9 ± 8.2 ^bc^
Range Somali (min–max)	-	1.1–2.1	3.3–5.1	13.4–20.2	75.9–91.1	4.3–14.1	3.9–15.2	0.2–1.3	22.4–57.4

* This sample is actually oval shaped, with a length of 55.0 ± 1.1 cm and a largeness of 46.2 ± 1.2 cm; n.d. = not determined. Data are presented as means ± standard deviation of three samples. Values with different letters in the columns are significantly different at *p* ≤ 0.05.

**Table 5 foods-12-03050-t005:** Colors of flours and breads.

Sample	*L**	*a**	*b**
Flours			
A *	70.96 ± 0.11 ^c^	3.48 ± 0.01 ^a^	9.23 ± 0.04 ^b^
E	83.40 ± 0.78 ^a^	0.13 ± 0.04 ^d^	7.72 ± 0.06 ^c^
F	78.39 ± 0.66 ^b^	1.13 ± 0.14 ^c^	9.36 ± 0.73 ^b^
G	80.11 ± 0.14 ^ab^	1.42 ± 0.02 ^b^	11.25 ± 0.01 ^a^
I	77.36 ± 0.08 ^b^	1.02 ± 0.01 ^c^	11.19 ± 0.01 ^a^
Breads—Bottom side			
A	38.25 ± 3.05 ^cd^	12.44 ± 1.00 ^bc^	24.33 ± 2.03 ^ab^
B	37.84 ± 2.02 ^cde^	14.57 ± 0.64 ^a^	19.74 ± 1.97 ^cd^
C	34.60 ± 2.53 ^e^	11.88 ± 0.67 ^cd^	22.49 ± 1.21 ^bc^
D	41.24 ± 2.81 ^b^	11.41 ± 1.06 ^d^	23.19 ± 1.75 ^b^
E	38.06 ± 3.63 ^cd^	12.98 ± 0.64 ^b^	22.59 ± 2.42 ^b^
F	39.93 ± 3.21 ^bc^	10.36 ± 0.51 ^e^	23.74 ± 1.24 ^b^
G	52.40 ± 2.53 ^a^	7.84 ± 0.70 ^g^	24.18 ± 1.85 ^ab^
H	51.22 ± 4.07 ^a^	9.85 ± 1.96 ^ef^	25.87 ± 3.37 ^a^
I	34.94 ± 2.39 ^de^	9.31 ± 0.95 ^f^	17.71 ± 0.66 ^d^
Breads—Superior side, clear areas			
A	47.52 ± 3.39 ^cd^	8.32 ± 0.43 ^b^	18.91 ± 1.18 ^c^
B	56.80 ± 1.73 ^a^	6.14 ± 0.68 ^ef^	25.73 ± 1.36 ^a^
C	34.12 ± 0.41 ^f^	6.72 ± 0.19 ^def^	18.02 ± 0.41 ^cde^
D	50.88 ± 2.96 ^b^	8.96 ± 1.00 ^a^	22.17 ± 1.14 ^b^
E	48.39 ± 2.31 ^bc^	7.74 ± 0.51 ^bc^	16.89 ± 0.93 ^e^
F	40.41 ± 4.12 ^e^	6.86 ± 0.8 ^de^	16.36 ± 2.54 ^e^
G	44.68 ± 3.76 ^d^	6.19 ± 0.52 ^f^	18.53 ± 1.56 ^cd^
H	50.71 ± 2.54 ^b^	6.31 ± 0.69 ^ef^	17.38 ± 1.77 ^de^
I	32.92 ± 1.99 ^f^	7.13 ± 0.45 ^cd^	14.06 ± 0.95 ^f^
Range: Somali (min–max)	32.92–50.88	6.19–8.96	14.06–22.17
Breads—Superior side, dark pattern (spiral or strip)			
A	42.94 ± 3.28 ^d^	8.55 ± 0.77 ^b^	19.07 ± 1.45 ^bc^
B	37.23 ± 1.43 ^e^	14.8 ± 0.46 ^a^	19.87 ± 2.03 ^abc^
C	-	-	-
D	47.41 ± 2.44 ^b^	9.13 ± 0.71 ^b^	21.36 ± 1.18 ^a^
E	46.13 ± 2.87 ^cd^	7.15 ± 0.65 ^cd^	15.14 ± 1.15 ^e^
F	39.63 ± 2.29 ^e^	7.38 ± 0.72 ^c^	17.41 ± 2.19 ^d^
G	46.63 ± 3.34 ^bc^	6.72 ± 0.80 ^de^	19.79 ± 0.78 ^b^
H	50.51 ± 1.86 ^a^	6.19 ± 0.82 ^e^	18.15 ± 1.35 ^cd^
I	27.78 ± 0.76 ^f^	3.59 ± 0.25 ^f^	5.42 ± 0.86 ^f^
Range: Somali (min–max)	27.78–50.51	3.59–9.13	5.42–21.36

* Sample of commercial *budo* composed of red sorghum without the addition of refined wheat flour. Breads, A, D, E, F, G, H, and I = samples of Somali *laxoox* bread; B and C = samples of Yemeni *lahoh* bread. Data are presented as means ± standard deviation of three samples. Different letters in a column for the same group of data indicate statistical differences at *p ≤* 0.05.

## Data Availability

All data are reported in the paper.

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
