# Peer review of "Physical-Chemical and Nutritional Characterization of Somali Laxoox Flatbread and Comparison with Yemeni Lahoh Flatbread"

_foods, 2023, doi:10.3390/foods12163050_

Round 1
Reviewer 1 Report
Kindly reveiw the editing and grammer, ome values marked yellow need revision, plgarism and similatry need to chick because you have published research papers in same trend and adea

Quality needs reveiw
Author Response
Reviewer 1
Dear Reviewer, we only found some yellow highlighting in the pdf file of the manuscript, with no comments. So, to answer we report here below the highlighted parts.
Line 6. Department of Soil, Plant and Food Science (DISSPA).
Answer: “DISSPA” is the acronym of the Italian wording. We deleted it to avoid confusion.
Line 17. Bioactives (2.44-12.51 mg/kg b-carotene, 5.02-7.11 mg/g gallic acid).
Answer: The sentence has been revised to be clearer and more precise (see lines 17-18).
Caption of Table 1. “….and other ingredients added to prepare the sampled Somali flatbreads”.
Answer: We reworded the caption of Table 1 to “…and other ingredients used to prepare the samples of Somali flatbreads”.
Table 3. Figures in this table.
Answer: It is not clear what the suggestion of the Reviewer could be in highlighting figures in this Table. We checked them all and they are ok, and were properly commented in the text. We only found that the values of antioxidant activity could be commented further, so these values have been commented in comparison with breads reported in other studies (see lines 392-394). In case the Reviewer does not agree with this modification, we kindly ask to be clearer in his/her suggestion.
Reviewer 2 Report
Even though the authors have tested a lot of parameters on flat breads from various households, the whole study is not scientific from the start (sample). Control variate method is the basis for scientific study design and effective comparasion and discussion. The samples from various house holds are processed by different people in different processing methods and different environments. Hence, the comparasion of the flat breads if meaningless. The reader can't get any effective hint from the study.
Author Response
Reviewer 2
Even though the authors have tested a lot of parameters on flat breads from various households, the whole study is not scientific from the start (sample). Control variate method is the basis for scientific study design and effective comparasion and discussion. The samples from various house holds are processed by different people in different processing methods and different environments. Hence, the comparasion of the flat breads if meaningless. The reader can't get any effective hint from the study.
Answer: Dear Reviewer, we are aware that to define the effect of a specific variable on the quality of food is important to follow the control variate method, but we wish to clarify that the aim of our study was not to define the effect of a specific variable on the quality of the end-product. Somali and Yemeni flatbreads have not been studied so far, so our research was aimed at giving a first characterization of their main physic-chemical and nutritional features and, considering that these flatbreads are artisanal products with a strong manual input in their preparation, it was fundamental to assess the wideness of the variability of these characteristics at different producers, by collecting bread samples processed by different people, with slight differences in the processing methods, and with different raw materials. Being the production process and the ingredients known, differences among breads could be reasonably examined, the ingredients appearing to be the major cause of variability. Anyway, these flatbreads are still not standardized, and at this stage is important to provide an insight into their nutritional characteristics and to assess their variability. The improvement of the nutritional quality of these breads, according to the control variate method, could be the object of further studies.
Moreover, please consider that “characterization” studies are quite common in scientific literature, as preparatory studies for further deepening, and these studies require considering real (commercial) food samples. See for example:
Sandvik, P., Marklinder, I., Nydahl, M., NÆs, T. and Kihlberg, I., 2016. Characterization of commercial rye bread based on sensory properties, fluidity index and chemical acidity. Journal of Sensory Studies, 31(4), pp.283-295;
Montazeri, N., Oliveira, A.C., Himelbloom, B.H., Leigh, M.B. and Crapo, C.A., 2013. Chemical characterization of commercial liquid smoke products. Food Science & Nutrition, 1(1), pp.102-115;
Zhao, Z., Mu, T. and Sun, H., 2019. Microbial characterization of five Chinese traditional sourdoughs by high-throughput sequencing and their impact on the quality of potato steamed bread. Food Chemistry, 274, pp.710-717;
Giannone, V., Giarnetti, M., Spina, A., Todaro, A., Pecorino, B., Summo, C., Caponio, F., Paradiso, V.M. and Pasqualone, A., 2018. Physico-chemical properties and sensory profile of durum wheat Dittaino PDO (Protected Designation of Origin) bread and quality of re-milled semolina used for its production. Food Chemistry, 241, pp.242-249.
Reviewer 3 Report
I found the article original and very interesting. It has some limitations in the research plan but they have been discussed by the authors and are reasonable. Below I submit some comments on the different sections of the article.
Abstract
line 17- you can't write that there were x mg of gallic acid in the flatbreads because it wasn't determined. I think you meant the total phenolic compounds expressed as mg/g GAE d.m.. Please verify and correct.
Material and methods
What is missing is a general description of how such bread is homemade (including dough making, baking). What are starter cultures used for? What is the process of making a flatbread in Yemen versus Somalia households? Are there similarities/differences?
Statistical analysis
Please provide in the text more information about the PCA analysis. Did all determinations enter as data for PCA analysis? If so, on what basis? What was the VIP score threshold? How many principal components were generated and what % of the total variability they explained (you may want to provide a table or a scatter plot).
In the statistics section or at the end of the introduction, it would be appropriate to write why PCA was chosen as the statistical analysis for comparing flatbreads of different origins and cite the utility of the tool for such purposes. Here I provide literature: https://doi.org/10.3390/antiox11112178 ; https://doi.org/10.3390/molecules27020355.
Results and discussion
I find all tables and charts necessary and clearly executed. The presentation and discussion of the results is appropriate and clear, well thought out.
Conclusions
Apart from the negative effect of mixing wheat flour with sorghum flour, the authors did not comment on the nutritional value of the breads tested. It may be worth comparing them with the composition of European bread to determine if they are nutritionally better/worse overall. The previously suggested literature can be used for this. Besides, the conclusions are original and come from the authors' research.
I found a few stylistic errors in several places in the text. Please reread the text and correct them.
Author Response
Reviewer 3
Comments and Suggestions for Authors
I found the article original and very interesting. It has some limitations in the research plan but they have been discussed by the authors and are reasonable. Below I submit some comments on the different sections of the article.
Answer: We thank the Reviewer for appreciating our work and for understanding the limitations of the sampling plan regarding the Yemeni breads, due to security challenges that hindered sample collection in Yemen.
Abstract
line 17- you can't write that there were x mg of gallic acid in the flatbreads because it wasn't determined. I think you meant the total phenolic compounds expressed as mg/g GAE d.m.. Please verify and correct.
Answer: The sentence has been amended (see lines 17-18).
Material and methods
What is missing is a general description of how such bread is homemade (including dough making, baking). What are starter cultures used for?
Answer: A detailed study of the production process of Somali flatbread has been the object of a previous work of the same authors (Wolgamuth, E., Yusuf, S., Hussein, A. and Pasqualone, A., 2022. A survey of laxoox/canjeero, a traditional Somali flatbread: production styles. Journal of Ethnic Foods, 9(1), 1-20, https://doi.org/10.1186/s42779-022-00138-3 ). Therefore, a brief description of the production process of Somali flatbread has been added in Materials and methods by quoting that paper for more detailed information (see lines 92-97).
What is the process of making a flatbread in Yemen versus Somalia households? Are there similarities/differences?
Answer: A detailed comparison of the breadmaking process in Yemen versus Somalia households has been added (see lines 555-573).
Statistical analysis
Please provide in the text more information about the PCA analysis. Did all determinations enter as data for PCA analysis? If so, on what basis? What was the VIP score threshold? How many principal components were generated and what % of the total variability they explained (you may want to provide a table or a scatter plot).
Answer: All the available determinations were used for the PCA, except "Thickness in spiral (mm)" and "Total anthocyanins", because those parameters were not available for all the samples. Being the variables different in their scales, PCA was carried out on the autoscaled data (i.e., on the correlation matrix) (Bro, R., & Smilde, A. K. (2014). Principal component analysis. Analytical methods, 6, 2812-2831). These information have been added in the text (lines 168-172).
Please consider that PCA is a multivariate unsupervised pattern recognition technique (Bro, R., & Smilde, A. K. (2014). Principal component analysis. Analytical methods, 6, 2812-2831; Li Vigni, M., Durante, C., & Cocchi, M. (2013). Exploratory data analysis. In Data handling in science and technology, Vol. 28, pp. 55-126, Elsevier) and it has been used in this way in the present work. Thus, the Variable Importance in Projection (VIP) were not calculated. Please, consider that such indices are very important for variables selection in supervised contexts, for example in regression problems (Westad, F., Bevilacqua, M., & Marini, F. (2013). Regression. In Data handling in science and technology, Vol. 28, pp. 127-170, Elsevier), while here our aim was to explore the structure of the dataset in an unsupervised manner. Thus, as suggested by Li Vigni, et al. (Li Vigni, M., Durante, C., & Cocchi, M. (2013). Exploratory data analysis. In Data handling in science and technology, Vol. 28, pp. 55-126, Elsevier), to the sole aim of data exploration, no formal/mathematical assessment of variables importance was carried out, while the discussion was driven by visual interpretation of scores and loadings. The first two PCs explained 58.45% of the variability in the data, as reported in the text (lines 530-531). Other PCs were calculated but not reported because our aim was to highlight the greatest differences/similarities among variables/samples and, by definition (Bro, R., & Smilde, A. K. (2014). Principal component analysis. Analytical methods, 6, 2812-2831), this is found in the first PCs.
In the statistics section or at the end of the introduction, it would be appropriate to write why PCA was chosen as the statistical analysis for comparing flatbreads of different origins and cite the utility of the tool for such purposes. Here I provide literature: https://doi.org/10.3390/antiox11112178 ; https://doi.org/10.3390/molecules27020355.
Answer: A paragraph was added in the statistics section (lines 168-172), but the suggested references were cited other parts of the manuscript, more properly (see refs. 25 and 47).
Results and discussion
I find all tables and charts necessary and clearly executed. The presentation and discussion of the results is appropriate and clear, well thought out.
Answer: We thank the Reviewer for appreciating our work.
Conclusions
Apart from the negative effect of mixing wheat flour with sorghum flour, the authors did not comment on the nutritional value of the breads tested. It may be worth comparing them with the composition of European bread to determine if they are nutritionally better/worse overall. The previously suggested literature can be used for this. Besides, the conclusions are original and come from the authors' research.
Answer: Thanks for suggesting. A comment on the nutritional value of the breads tested has been added (lines 276-295).
Comments on the Quality of English Language
I found a few stylistic errors in several places in the text. Please reread the text and correct them.
Answer: Thanks for noting, we carefully reread and amended several stylistic errors.
Reviewer 4 Report
The manuscript ID: foods-2520064 entitled ''Physical-chemical and nutritional characterization of Somali laxoox flatbread and comparison with Yemeni lahoh flatbread" presents a very interesting topic, of scientific sounds. This manuscript is well structured and written.
The Introduction section provide background about the topic.
The experimental design is adequately discussed.
Interesting results were obtained by suitable methods and data interpretation is robust, valid and reliable.
The Conclusion section was supported by the results, was well written and provides a good conclusion for the study.
Author Response
Reviewer 4
The manuscript ID: foods-2520064 entitled ''Physical-chemical and nutritional characterization of Somali laxoox flatbread and comparison with Yemeni lahoh flatbread" presents a very interesting topic, of scientific sounds. This manuscript is well structured and written.
The Introduction section provide background about the topic.
The experimental design is adequately discussed.
Interesting results were obtained by suitable methods and data interpretation is robust, valid and reliable.
The Conclusion section was supported by the results, was well written and provides a good conclusion for the study.
Answer: We thank the Reviewer for appreciating our work.
Reviewer 5 Report
This research aims to study the physical-chemical and nutritional properties of Somali laxoox flatbread and draw a comparison with Yemeni lahoh flatbread.
This topic has remained unexplored to date, making this study novel. The data collection involved obtaining samples directly from households and employing manual preparation methods. However, it is essential to acknowledge that this approach may introduce some variability in the results and limit the representativeness of the bread characteristics pertaining to the culinary traditions and cultures of Somalia and Yemen. Thus, it is pertinent to clarify that the samples analyzed for physical-chemical and nutritional properties comprise a mixture of various bread samples collected (example: 27 to Somali laxoox).
The results of this investigation have shed light on the physical-chemical and nutritional properties of Somali laxoox flatbread, providing insights into its composition and nutritional profile.
The work contributes to the knowledge of the physical-chemical and nutritional attributes of flatbread and offered valuable insights into its cultural, historical, and ethnographic significance.
Minor comments:
- Change the notation of nutritional and bioactive compounds. Example: use "12.47 g proteins/100 g" instead of "12.47g/100 g proteins," or "5.02 mg gallic acid/g" instead of "5.02 mg/g gallic acid."
English language fine.
Author Response
Reviewer 5
This research aims to study the physical-chemical and nutritional properties of Somali laxoox flatbread and draw a comparison with Yemeni lahoh flatbread.
This topic has remained unexplored to date, making this study novel. The data collection involved obtaining samples directly from households and employing manual preparation methods. However, it is essential to acknowledge that this approach may introduce some variability in the results and limit the representativeness of the bread characteristics pertaining to the culinary traditions and cultures of Somalia and Yemen. Thus, it is pertinent to clarify that the samples analyzed for physical-chemical and nutritional properties comprise a mixture of various bread samples collected (example: 27 to Somali laxoox).
Answer: A sentence to clarify the sampling plan has been added at the beginning of the Results and discussion section.
The results of this investigation have shed light on the physical-chemical and nutritional properties of Somali laxoox flatbread, providing insights into its composition and nutritional profile. The work contributes to the knowledge of the physical-chemical and nutritional attributes of flatbread and offered valuable insights into its cultural, historical, and ethnographic significance.
Answer: We thank the Reviewer for appreciating our work.
Minor comments:
- Change the notation of nutritional and bioactive compounds. Example: use "12.47 g proteins/100 g" instead of "12.47g/100 g proteins," or "5.02 mg gallic acid/g" instead of "5.02 mg/g gallic acid."
Answer: The notation has been changed throughout the entire text and tables.
Round 2
Reviewer 2 Report
The authors have carefully revised the manuscript, the revised version is acceptable.
Author Response
Reviewer 2
The authors have carefully revised the manuscript, the revised version is acceptable.
Answer: We thank the Reviewer for appreciating our revision.
Reviewer 3 Report
The authors addressed all amendments and comments.
They made all the additions to the missing information in the methodology and statistics section. They also expanded the discussion and conclusions section. The authors have also improved the tables of results making them even more readable.
Author Response
Reviewer 3
The authors addressed all amendments and comments.
They made all the additions to the missing information in the methodology and statistics section. They also expanded the discussion and conclusions section. The authors have also improved the tables of results making them even more readable.
Answer: We thank the Reviewer for appreciating our revision.